# Systematic Review of IoT-Based Solutions for User Tracking: Towards Smarter Lifestyle, Wellness and Health Management

**DOI:** 10.3390/s24185939

**Published:** 2024-09-13

**Authors:** Reza Amini Gougeh, Zeljko Zilic

**Affiliations:** Faculty of Engineering, Department of Electrical and Computer Engineering, McGill University, Montreal, QC H3A 0G4, Canada; zeljko.zilic@mcgill.ca

**Keywords:** Internet of things (IoT), health monitoring, machine-to-machine (M2M) interaction, wearable sensors, human–computer interaction (HCI), real-time monitoring, health and wellness management

## Abstract

The Internet of Things (IoT) base has grown to over 20 billion devices currently operational worldwide. As they greatly extend the applicability and use of biosensors, IoT developments are transformative. Recent studies show that IoT, coupled with advanced communication frameworks, such as machine-to-machine (M2M) interactions, can lead to (1) improved efficiency in data exchange, (2) accurate and timely health monitoring, and (3) enhanced user engagement and compliance through advancements in human–computer interaction. This systematic review of the 19 most relevant studies examines the potential of IoT in health and lifestyle management by conducting detailed analyses and quality assessments of each study. Findings indicate that IoT-based systems effectively monitor various health parameters using biosensors, facilitate real-time feedback, and support personalized health recommendations. Key limitations include small sample sizes, insufficient security measures, practical issues with wearable sensors, and reliance on internet connectivity in areas with poor network infrastructure. The reviewed studies demonstrated innovative applications of IoT, focusing on M2M interactions, edge devices, multimodality health monitoring, intelligent decision-making, and automated health management systems. These insights offer valuable recommendations for optimizing IoT technologies in health and wellness management.

## 1. Introduction

Over the past decade, the Internet of Things (IoT) has experienced exponential growth, with estimates suggesting that more than 20 billion IoT devices are currently operational worldwide [1]. Multiple devices, interconnected across various layers of IoT systems, autonomously process events and transmit data without human interaction [2]. This technological paradigm is transformative across diverse sectors, improving urban management and industrial operations while also extending its reach to unexplored domains, such as deep-sea and space technologies [3,4,5,6]. In the healthcare sector, IoT offers unprecedented opportunities for enhancing health, wellness, and lifestyle management, particularly through its integration with biosensor technologies [7]. Reflecting this potential, the global IoT healthcare market size reached USD 44.21 billion in 2023 and is expected to expand at a compound annual growth rate of 21.2% from 2024 to 2030 [8].

Biosensors are made tremendously more useful when applied in IoT applications. The IoT connectivity enables the direct measurement of physiological parameters and provides real-time, accurate, and continuous health data. The integration of biosensors for humans with IoT results in the development of advanced smart health devices. These devices collect, transmit, and analyze biometric data in real-time [9]. In fact, IoT-enabled biosensors demonstrate versatility in various medical applications; for example, they can monitor glucose levels in diabetic patients with high precision, track heart rate variability to detect early signs of cardiac issues, and measure stress levels through cortisol detection [10,11,12].

Despite these advances in health and wellness tracking technologies, many individuals still struggle with effective lifestyle management [13]. This challenge is compounded by several critical issues in IoT healthcare implementations. Data security remains a concern, necessitating advanced encryption methods, like the Advanced Encryption Standard (AES), and secure protocols, such as transport layer security (TLS), to protect sensitive health information [14]. User privacy is equally crucial, with regulations like the Health Insurance Portability And Accountability Act (HIPAA) in the US and the General Data Protection Regulation (GDPR) in Europe guiding the ethical collection and use of personal health data [15]. Device interoperability poses another significant challenge, as the lack of standardization hinders seamless integration. Initiatives like OneM2M and the Open Connectivity Foundation (OCF) are working to develop universal standards to address this issue [16]. Finally, ensuring the long-term usability of IoT devices is essential for continuous health monitoring, requiring solutions for extended battery life, regular firmware updates, and adaptability to evolving health needs [17].

In addition to these challenges, current systems often fail to integrate data modalities across different aspects of health and wellness. For instance, some systems focus solely on inertial measurement unit (IMU) sensors to monitor physical activity or electrocardiogram (ECG) sensors to track heart health, without integrating these data with other critical health indicators, such as nutritional intake or mental well-being [18,19]. The unimodal approach limits the ability to provide comprehensive health insights and personalized recommendations. However, advances in computational power and IoT technologies are driving a shift towards the development of multimodal systems [20,21]. For example, researchers have recently proposed systems that combine information received from different biosignals and sensor inputs to track users and offer a more in-depth view of health and wellness [22,23,24].

Novel IoT systems facilitate multimodal interactions through advanced frameworks such as machine-to-machine (M2M) communications. Their primary goal is to enable an efficient data exchange between devices without human intervention [25,26]. These frameworks utilize protocols such as message queuing telemetry transport (MQTT) and constrained application protocol (CoAP) for lightweight and efficient communication [27]. Cloud processing capabilities enhance this integration by allowing vast amounts of data to be processed and analyzed quickly [28]. Concurrently, edge computing is gaining prominence in IoT healthcare applications, reducing latency and improving real-time data processing. This approach allows for immediate analysis and decision-making at the device level, which is crucial for time-sensitive health monitoring scenarios [29]. On-device processing ensures that critical data can be handled locally for immediate feedback and action [30]. This is particularly important for energy efficiency, as it reduces the need for constant data transmission, thereby extending battery life in wearable IoT devices. Human–computer interaction (HCI) plays a vital role in this ecosystem. Intuitive mobile and desktop applications provide users with accessible interfaces to manage and interpret their health data effectively [31]. These applications not only display real-time data but also offer personalized recommendations and alerts, which empower individuals to make informed decisions about their health and wellness [32].

Despite the recognized impact of IoT on health and lifestyle management, current systematic reviews in this area often lack a comprehensive evaluation of the integration of M2M, HCI, and multimodal biosensing technologies within health and wellness management systems. While existing mappings typically focus on individual aspects, such as IoT sensor technologies or communication frameworks, they seldom assess the combined effects of these technologies on health management. This paper distinguishes itself by systematically comparing and analyzing the use of IoT-enabled biosensors, M2M communication, and HCI technologies, emphasizing their potential to deliver continuous, automated, and multimodal user tracking. By addressing both technological integration and HCI, this review offers a unique contribution to the field, identifying gaps and proposing advancements in the implementation of IoT systems for personalized health and lifestyle management. This study is structured around the following research questions:What types of biosignal sensors and IoT devices are predominantly used in M2M interactions, and how do these technologies compare in terms of their psychophysiological modalities and effectiveness in health monitoring?How are biosignal sensors and other IoT devices integrated into M2M systems for health and lifestyle management, and what roles do these components play within decision-making frameworks (rule-based, ML-based, or hybrid)?What key outcomes and advancements have been achieved through the use of IoT, particularly in the integration of HCI and ’human-to-many computers’ scenarios, and what potential future applications could advance health, wellness, and lifestyle management?

The remainder of this paper is organized as follows: Section 2 contains the methodology used in this systematic review. Section 3 presents the results, along with the limitations and recommendations for future studies. Finally, Section 4 concludes the paper.

## 2. Methodology

This systematic review was conducted according to the preferred reporting items for systematic reviews and meta-analyses (PRISMA) guidelines [33]. Moreover, the protocol for this study was registered with the Open Science Framework (OSF) Registries (available publicly at https://osf.io/v5kb3 (accessed on 7 September 2024)).

### 2.1. Search Strategy

In this work, we used four key databases: Scopus and Web of Science, which are comprehensive indexing services, alongside IEEE Xplore and the ACM Digital Library, which are specialized repositories for research in technology and engineering. These databases were chosen for their broad coverage of peer-reviewed articles in IoT, M2M interactions, and health monitoring since they provide a comprehensive and overlapping indexing of relevant journals. A search of journal papers published between January 2019 and December 2023 was conducted to capture the last five years of advancements. Keywords were grouped and combined as follows to focus on relevant studies while excluding less pertinent subjects (set1 AND set2 AND set3 AND NOT set4):Set1: (“machine-to-machine” OR m2m OR “d2d” OR “device-to-device” OR iomt OR iot).Set2: (wearable OR “personal health device” OR mobile OR “body area” OR sensor OR track* OR biosignal OR monitor*).Set3: (exercise OR “physical activity” OR diet OR health OR fitness).Set4: “air quality” OR “future directions” OR economic OR nano* OR agricultur* OR “data sharing” OR antenna OR trends OR challenges OR survey OR biochip OR “smart city” OR “smart building*” OR opportunities).

The term “physiological computing” was not used as a keyword as it is a very specialized term that few researchers in the M2M field utilize. Moreover, we included the term “IoT” since most of the studies may not use M2M or D2D interactions in their paper.

### 2.2. Study Selection Process

From the initial pool of 4313 articles, 28 were from ACM, 474 from IEEE Xplore, 1368 from Web of Science, and 2443 from Scopus. A flowchart of the study selection process is depicted in Figure 1. After removing duplicates, 2917 studies remained for screening based on their titles and abstracts. This initial screening excluded 2349 papers that did not meet the inclusion criteria, leaving 568 articles for full-text analysis. Ultimately, 19 articles focusing on IoT, M2M, and user tracking through wearables were included. The publication year analysis shows a peak in 2020 with 10 articles, followed by 2022 with 6 and 2019 with 5. The years 2021 and 2023 each contributed two articles. This distribution indicates a peak in research output in 2020, with decreased activity in 2021 and a resurgence in 2022, reflecting dynamic research trends in the field.

### 2.3. Inclusion and Exclusion Criteria

Inclusion criteria for this review include studies that: (1) Use wearable devices to measure physiological metrics, such as body temperature (BT), heart rate (HR), or blood pressure (BP), as well as body motion data from human subjects; (2) Explore the integration of these sensors in an M2M setting; (3) Discuss the system design, decision-making frameworks (rule-based, machine learning-based, or hybrid), and/or sensor integration strategies in terms of M2M/HMI/HCI frameworks; (4) Present empirical findings, advancements, or potential applications in the field of M2M or HMI/HCI using the captured data. Exclusion criteria include: (1) Review papers and theoretical articles without empirical data collection; (2) Studies that do not focus on wearables; (3) Papers that do not address the integration of these sensors into system design or decision-making processes in M2M or HMI/HCI contexts; and (4) Studies that only propose a theoretical framework or an algorithm for feature fusion and classification purposes.

Eligibility criteria according to population, intervention, comparison, outcome, and study design (PICOS) have been tabulated in Table A1 and are available in Appendix A.

### 2.4. Screening and Data Extraction

Following the search of the specified databases and the removal of duplicate entries, the authors conducted a preliminary screening of titles and abstracts to filter out irrelevant studies. The remaining articles underwent full-text screening, and those that met the criteria were included in the systematic review. A data extraction spreadsheet was created to gather detailed information from each study, covering three main areas: (1) study design and demographics (target group, participant count or dataset name, gender distribution, target issue, and experiment description), (2) methodological choices and implementation elements (equipment used, mobile or web apps developed, sensor modalities, system security), and (3) outcomes.

In addition to these domains, further areas were considered to enhance the comprehensiveness of the review: (4) user interaction and experience (usability, engagement, accessibility, and user satisfaction), (5) data analytics and interpretation (methods of data analysis, visualization techniques, and personalized recommendations), (6) ethical and privacy considerations (ethical issues, privacy policies, and data security), (7) interoperability and integration (compatibility with other systems, integration with healthcare infrastructure, and compliance with standards), (8) long-term effectiveness and sustainability (impact on health, technology maintenance, and cost-effectiveness), and (9) context and environment (usage settings, environmental factors, and adaptability to different cultural and socioeconomic contexts).

### 2.5. Synthesis Methods

For synthesis, studies were categorized based on their focus areas (e.g., health monitoring, fitness tracking), technologies employed (e.g., wearables, IoT devices), and target populations (e.g., elderly, patients with chronic conditions). This structured approach ensured that similar studies were grouped for meaningful comparison and synthesis.

During the data preparation, standardization methods were applied to address variations in the reporting of performance metrics, such as classification accuracy, root-mean-square errors (RMSE), and usability scores. For example, the high variation in accuracy across different classifiers (e.g., random forest) was managed by transforming measures to a common scale where possible. Missing statistics, such as standard deviations or confidence intervals, were imputed following the established guidelines, while effect sizes or standardized mean differences were used to convert odds ratios where required.

For clarity in presenting the synthesis, we employed visual tools, such as alluvial diagrams, to summarize our findings from the included studies. Data were transformed where needed to standardize diverse outcome measures. For example, exercise recognition accuracy, health monitoring usability, and fall detection performance were synthesized into comparable metrics to allow for cohesive analysis across different sensor modalities and study designs.

### 2.6. Quality Assessment

We used the assessment tool STROBE (STrengthening the Reporting of Observational Studies in Epidemiology) [34], which applies 22 checks covering the title and abstract (item 1), introduction (items 2 and 3), methods (items 4–12), results (items 13–17), discussion (items 18–21), and funding information (item 22). Eighteen items apply to all observational studies (OSs), while four items have subcategories specific to certain types of OSs [34]. To facilitate scoring, we categorized some sub-items into 53 smaller items (e.g., item No. 5 becomes 5a, 5b, 5c, 5d) with equal weights. Each item mentioned in the appropriate section of the article received one point; otherwise, it received zero points. Items not applicable to a specific article were not considered. We then calculated compliance rates by averaging the scores across all studies for each STROBE item and across all STROBE items for each study to determine adherence levels.

## 3. Results and Discussion

In this section, we provide the findings from the collected data, focusing on the research questions outlined in the introduction.

### 3.1. Study Design and Characteristics

Detailed information about the configuration, equipment, study methodology, experiment description, and demographics of the participants is listed in Table 1. The description of equipment, software, and the placement of wearables is given in Table 2.

From the reviewed articles, one study targeted individuals with incorrect body posture [35], while two focused on patients diagnosed with diabetes [36,37]. Eight studies only recruited healthy participants [38,39,40,41,42,43,44,45]. Additionally, three studies gathered data from clinical patients [46,47,48], and four studies concentrated on elderly populations [49,50,51,52]. Furthermore, one study did not specify the population source [53]. The reported numbers in Table 1 represent participants who completed all sessions and adhered to all experimental protocols. Overall, the sample sizes were above five participants, with age ranges varying from seven to 78 years.

According to Table 2, the reviewed studies employed various sensors and devices to monitor and collect data for health and fitness assessments. Two articles focused on fitness and health activities [35,38], while two other studies concentrated on health monitoring [36,46]. Notably, various hardware platforms, such as wristwatch sensors, IMU sensors, and digital recording devices, were utilized. Among the included articles, six studies used more than one type of sensor to collect data [35,36,48,49,50,51].

**Table 1 sensors-24-05939-t001:** Demographic and experimental details of the included studies.

Study	Target Population	Target Variable	Experiment	Population	Age Range	F/M	Duration
[35]	Wrong body posture	Muscle movements, posture during exercises	T-bar and bicep curl exercises with smart fitness suite	9	20–10	NA	2 h daily
[36]	Type 1 diabetes patients	Glucose levels 1 h post-exercise	Devices and sensors for glucose and physical activity data	26	7–15	NA	5–6 days
[38]	Healthy people	Pressure distribution on hand palm during exercises	Fitness tracking with a smart glove	10	22–30 (25 ± 3.21)	1/10	1 h session
[46]	Clinical patients	SpO_2_, heart rate	Wristwatch sensor for SpO_2_ and heart rate in clinical trial	24	NA	NA	4 weeks
[49]	Elderly with MCI	HR, BP, activities	Monitoring health of MCI patients	50	78 (avg)	25/25	NA
[50]	Elderly	Various physiological parameters	System usability with physiological monitoring	5	60–65	2/3	-
[51]	Nursing home elderly	BP, glucose, HR, temp, oxygenation	Healthcare monitoring in nursing homes	5	NA	4/1	One time
[39]	Healthy people	Toe-out angle	Smart shoes for gait pattern analysis	11	27.8 ± 5.1	2/9	NA
[53]	Patients requiring continuous blood pressure monitoring	BP, HR, motion data	Data collection on BP, heart rate, motion for calibration	NA	NA	NA	10 min/trial, 65 trials total
[52]	Elderly	Physio data (exercise, diet) and environmental factors (temp, air quality)	Health promotion system feedback	30	62–68	14/16	-
[48]	Asthma patients	Temp, HR, O_2_, room temp, humidity, air quality	Continuous monitoring using IoT sensors	15	NA	6/9	-
[37]	Diabetes patients	HbA1c levels	Bluetooth devices for health metrics, bi-monthly guidance	49	54.4 ± 6.0	All male	6 months
[47]	Patients needing continuous monitoring	Sensor data integrity	ECG, SpO_2_, temp sensors with Arduino, smartphone analysis	5	NA	NA	10 s interval, 2 ms/sample, 5000 samples/person
[42]	Gym-goers	Exercise and health data	Workout regimen with wearable tech	4	NA	All male	6 weeks, 6 days/week
[40]	Healthy (elderly for final)	Fall events, heart rate	Simulating falls and ADL for system testing	5	23–24	NA	125 fall trials, 225 ADL trials
[43]	College students	Autism quotient	Multisensor wearable, data on audio, motion, environment, psychological states via questionnaires	16	22 (avg)	1/15	1 month, daily data collection
[44]	Athletes in head-impact sports	Strength of head hits	Boxing helmet test in realistic matches	6	NA	NA	3 matches
[45]	General population	COVID-19 symptoms, social distancing	Smart COVID-Shield deployment at aluminum factory	678–835	>60, <20	NA	9 weeks
[41]	Healthy people	Physical activities	Data from sensors on wrists, arms for 13 activities, smartphone and web-server analysis	19	27 ± 8	NA	11 h total

Acronyms: F/M: female/male, BP: blood pressure, HR: heart rate, SpO_2_: oxygen saturation, ECG: electrocardiogram, MCI: mild cognitive impairment, ADL: activities of daily living, HbA1c: hemoglobin A1c, NA: not available.

**Table 2 sensors-24-05939-t002:** Hardware and software characteristics of included studies.

Study	Hardware	Location	Software
Equipment	Processor	Data Types	Development Environment	App
[35]	IMU (MPU 6500) ^1^, EMG (MyoWare) ^2^, Wi-Fi (NodeMCU Esp 8266 v3) ^3^	Arduino Mega 2560 ^4^, Cloud-layer	Upper back	Physio, sensor data	Java	
[36]	Enlite sensor (MMT-7008A) ^5^, Xiaomi Mi Band 5 ^6^	CGM (iPro2 MT-7745WW) ^5^, CareLink iPro web ^5^	Wrist	Physio, sensor data	Xamarin	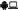
[38]	FSR (Interlink 402) ^7^, Adafruit Feather 32u4 Bluefruit ^8^, 16 channel Multiplexer	Local computer	Forearm, hand	Sensor data	MATLAB (NA version)	
[46]	BLE module (nRF52) ^9^	SAM R30 Xplained Pro ^10^	Wrist	Physio data	Java	
[49]	Withings MoveECG ^11^, BP monitor, scale, sleep analyzer, Wi-Fi (Node MCU) ^3^	Raspberry Pi ^12^, Cloud-layer	Leg	Physio, sensor data	NA	NA
[50]	MySignals HW V2 ^13^, Arduino UNO ^4^, Wi-Fi (ESP8266) ^3^	Cloud-layer	NA	Physio, sensor data	C++, React Native, MongoDB, Node.js	
[51]	Hexiwear ^14^, Galaxy S7 ^15^	Cloud-layer (WolkAbout platform) ^16^	Wrist	Physio data	Node.js, React Native, MongoDB	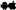
[39]	Accelerometer Gyroscope (MPU 6050 sensor) ^1^	Local computer	Feet	Sensor values	NA	NA
[53]	MKB0803 HR and BP module ^17^, LPMS-ME1 DK IMU ^18^, OMRON U30 sphygmomanometer ^19^	Raspberry Pi 3 B+ ^12^	Upper arm	Physio data	Python, JS	
[52]	Wearable sensors, Homecare server, Temp, humidity, particulate matter 2.5 sensors	Mobile app	NA	Physio, environment data	MySQL, Java	
[48]	Thermometer (DS18B20) ^20^, Pulse-oximetry (MAX30100) ^20^, Humidity (DHT-11) ^8^, Dust (GP2Y1010) ^21^, Air quality (MQ-135) ^22^, LCD Display	Arduino Uno ^4^, Wi-Fi (Node MCU ESP8266) ^3^, Cloud-layer	NA	Physio, environment data	Arduino IDE, Java	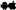
[37]	TOSHIBA Actiband WERAM1100 ^23^, BP (UA-851PBT-C) ^24^, Digital scale (UC-411PBT-C) ^24^	Cloud-layer	Wrist	Physio data	NA	
[47]	MySignals HW V2 ^13^, Arduino UNO R3 ^4^, ECG, SpO_2_, Temp sensors	Cloud-layer	Waist, chest	Physio, sensor data	MATLAB (7.12.0.635, R2011a)	
[42]	Zephyr BioHarness 3 ^25^	Fog-layer	Chest	Physio, sensor data	Java	
[40]	Accelerometer, Bluetooth (HC-05) ^26^, Pulse sensor, Lithium-ion battery ^21^	Atmega 328 p ^10^	Waist	Physio, sensor data	Java, Google firebase	
[43]	Wearable device, OLED, SD card, Bluetooth	Micro-controller (STM32F405) ^27^	Wrist	Physio, sensor, audio, questionnaire	Java	
[44]	IoT helmet, Temperature (LM35) ^28^, Accelerometer (GY-61 ADXL335) ^29^, Wi-Fi (ESP8266) ^3^	Arduino Nano ^4^	Helmet	NA	C++, JS, PHP, MySQL, Arduino IDE	
[45]	PIR sensor, Temp, Ultrasonic sensors, Belt, Suspender	Arduino UNO ^4^	Waist, shoulders	Physio, sensor data	Arduino IDE (v2.0 beta)	NA
[41]	MetaMotion R64 ^30^, Smartphones, Web-server	Local computer, Cloud-layer	Wrist, arms	Physio, sensor data	PHP, Python, MySQL	

Acronyms: NA: not applicable, IMU: inertial measurement unit, EMG: electromyography, CGM: continuous glucose monitor, FSR: force-sensitive resistor, BLE: Bluetooth low energy, BP: blood pressure, HR: heart rate, PM: particulate matter, SpO_2_: oxygen saturation, ECG: electrocardiogram, GPS: Global Positioning System, PIR: passive infrared sensor, JS: JavaScript. Icons: 

 Android app, 
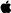
 iOS app, 

 web app. ^1^ InvenSense, San Jose, CA, USA; ^2^ Advancer Technologies, Raleigh, NC, USA; ^3^ Espressif Systems, Shanghai, China; ^4^ Arduino, Ivrea, Italy; ^5^ Medtronic, Minneapolis, MN, USA; ^6^ Xiaomi, Beijing, China; ^7^ Interlink Electronics, Camarillo, CA, USA; ^8^ Adafruit Industries, New York, NY, USA; ^9^ Nordic Semiconductor, Trondheim, Norway; ^10^ Microchip Technology Inc., Chandler, AZ, USA; ^11^ Withings, Issy-les-Moulineaux, France; ^12^ Raspberry Pi Foundation, Cambridge, ^13^ MySignals, Libelium, Zaragoza, Spain; ^14^ MikroElektronika, Belgrade, Serbia; ^15^ Samsung, Suwon, South Korea; ^16^ WolkAbout, Novi Sad, Serbia; ^17^ Yunnan Electric Power Materials, Yunnan, China; ^18^ LP-Research, Tokyo, Japan; ^19^ OMRON, Tokyo, Japan; ^20^ Maxim Integrated, Wilmington, MA, USA; ^21^ Socle Technology Corp., Hsin-Chu, Taiwan; ^22^ Winsen Electronics, Zhengzhou, China; ^23^ Toshiba, Tokyo, Japan; ^24^ A&D Company, Tokyo, Japan; ^25^ Zephyr Technology, Annapolis, MD, USA; ^26^ James Electronic, Belmont, CA, USA; ^27^ STMicroelectronics, Plan-les-Ouates, Switzerland; ^28^ Texas Instruments, Dallas, TX, USA; ^29^ Analog Devices, Wilmington, MA, USA; ^30^ MbientLab, San Jose, CA, USA.

### 3.2. Quality Assessment of Included Studies

STROBE was used to assess the quality of the included papers. The detailed responses obtained by the assessment tool are presented in Table A2 available in Appendix A. The “Title and Abstract” category shows high compliance, with most studies indicating their design and providing balanced summaries. The “Introduction” section shows perfect compliance, with all studies meeting the requirements [35,36,37,38,39,40,41,42,43,44,45,46,47,48,49,50,51,52,53]. The “Discussion” section has a compliance rate of 99%, with only one study not fully adhering to it [41]. The “Methods” section has an average compliance rate of 89%, indicating that some studies do not fully detail their methodological approaches [39,45,48,51,52,53]. The “Results” section also shows a similar compliance rate of 89%, suggesting that the reporting of results could be improved in several studies [39,45,48,51,52,53]. The study-specific analysis identifies [42,43,44] as highly compliant, with compliance rates close to or above 1.95. Conversely, one study stands out with a significantly lower compliance rate of around 1.06 [41].

### 3.3. Methodology and Implementation in Included Studies

Overall, the included articles utilized a variety of technologies. Figure 2 shows the breakdown of the methods. This diagram illustrates the flow from sensors and computational units through data transfer, processing, feature extraction, preprocessing, classification, and secure data transmission to user tracking, focusing on health, fitness, and wellness. The following subsections provide a detailed overview of these methods.

#### 3.3.1. Processing Platforms

The reviewed studies utilized various computing tools to process data gathered from sensors, including microcontrollers, single-board computers, cloud computing, and local computing platforms. Arduino (Arduino, Ivrea, Italy) was employed in three studies [35,44,45]. Other microcontroller platforms included Raspberry Pi (Raspberry Pi Foundation, Cambridge, UK) [53] and advanced RISC machine (ARM) microcontrollers (Microchip Technology Inc., Chandler, AZ, USA) [40,43]. Cloud computing, on the other hand, was used in six studies. Researchers leveraged cloud platforms to enhance their IoT applications, with real-time data analytics capabilities and remote health monitoring [36,37,49,50,51,52].

Three studies used local computing solutions to provide immediate data processing and real-time feedback [38,39,42]. Two studies integrated both mobile phones and cloud computing to balance real-time data collection with advanced cloud-based processing [41,47]. This hybrid approach highlights the flexibility and adaptability of computing tools in supporting diverse IoT-based health and fitness monitoring applications. One study employed an unspecified processor within a wearable device for data processing [46].

#### 3.3.2. Sensor Modalities

The studies incorporated a diverse range of sensors and sensing devices to monitor physiological and environmental data. Overall, 15 types of sensor modalities were used in the included studies. The following is a summary of the modalities and sensors used in these studies.

**IMU and Motion Sensors:** The reviewed studies extensively utilized IMU and motion sensors to capture and analyze user movement data. Five studies incorporated these essential components into health monitoring systems to gather detailed insights into physical activities, body movements, and postural changes [35,39,40,44,53]. These sensors provided information about muscle movements and posture during exercises to detect deviations that could lead to injuries [35]. In the analysis of gait patterns, smart shoes measured the toe-out angle, offering insights into walking mechanics [39]. For monitoring physiological parameters, the Raspberry Pi 3 B+, along with IMU sensors, tracked motion to ensure accurate data collection [53]. Accelerometers and other motion sensors gathered data on physical activity levels and detected falls, contributing to comprehensive health monitoring [40]. An IoT helmet with an IMU sensor monitored head movements and provided real-time feedback for safety and health assessments [44].

**Biosignal Monitoring:** Electrocardiography (ECG) sensors, which are essential in cardiac monitoring, are used to measure and record the electrical activity of the heart. They detect abnormalities in heart function and provide critical information for diagnosing and managing cardiac conditions [42,47,49]. In contrast, heart rate monitors, a fundamental feature in health tracking devices, measure the number of heartbeats per minute [36,42,46,48,49,50,51,53]. This modality offers insights into cardiovascular health and aids in fitness assessment and overall health monitoring. *Blood pressure monitors*, used in three studies, measure systolic and diastolic pressure. This information is crucial for diagnosing and treating hypertension [37,49,50,53].

Electromyography (EMG) sensors measure muscle activity by detecting electrical signals from muscle contractions. They assess muscle function and exercise effectiveness. One study used EMG sensors on the biceps brachii to monitor movements and ensure proper form and improve exercise and fitness routines [35]. *Respiration sensors* track breathing patterns and breath rate (BR), which is essential for assessing respiratory health and detecting anomalies during exercise. This real-time data aids in health monitoring and alerts trainers or healthcare professionals to potential respiratory issues [42].

*Pulse-oximeters* measure blood oxygen levels, which is crucial for monitoring respiratory function. They often use photoplethysmography (PPG) to measure blood oxygen saturation and heart rate [40,46,47,48,50,51]. *Glucometers* measure blood glucose levels, essential for managing diabetes. Two studies highlighted the importance of using glucometers for monitoring glucose levels in patients and emphasized the critical role of continuous glucose monitoring (CGM) in maintaining optimal health for diabetic individuals [36,50].

**Physical Activity Tracking:** Sensors that count steps and calories are essential for fitness monitoring. Among the reviewed studies, one study integrated wearable sensors with smartphones to enhance activity monitoring [36]. Another developed a platform to track elderly people’s activities [49]. A third study used sensors to monitor postural changes and movements to prevent musculoskeletal disorders [37]. Interlink *Force-sensitive resistor* (FSR) 402 sensors (Interlink Electronics, Camarillo, CA, USA) measure pressure and force for smart gloves that track hand movements and pressure distribution. One study utilized FSR sensors to gather user interactionsdata [38].

**Body Temperature Sensors:** Nine studies incorporated body temperature sensors for health monitoring and early detection of physiological changes. These sensors monitored elderly patients for early signs of fever and other health issues [49]. They detected abnormal temperature variations that indicated potential problems [50]. They were also used in wearable devices for remote monitoring of elderly patients [51]. In asthma management, sensors collected real-time data on patients’ health status using an IoT-based system [48]. A WBAN-based system collected body temperature data to gather comprehensive physiological information [47]. For athletes, sensors in IoT-based helmets monitored dangerous hits and elevated temperatures and alerted medical staff [44]. During the COVID-19 pandemic, these sensors continuously measured body temperature and notified users of fever as part of a smart tracking system [45]. They also played a role in analyzing physical activities within a web-based framework [41]. In smart gyms, sensors monitored athletes’ health and generated real-time alerts during workouts [42].

**Location Sensors and Environmental Measures:** GPS sensors provide location data and enhance the contextual understanding of health monitoring. One study used GPS to send the user’s location to caregivers via a mobile application for immediate help after detecting a fall [40]. Location sensors tracked the daily movements and activities of elderly patients to assess and predict their health status [49].

Environmental sensors monitor temperature, humidity, and air quality to analyze their relationship with the psychological and mental states of participants [43]. These sensors enhance the safety and quality of life for elderly individuals by providing context-aware services [49,52]. Ambient sensors identify asthma triggers and ensure a safe environment for asthmatic patients by monitoring these conditions [48].

#### 3.3.3. User Interfaces

The reviewed studies implemented a range of user interfaces (UIs) to facilitate interaction between users and IoT-based systems. The most common mediums included mobile applications, web-based interfaces, and cloud applications. Ten studies developed Android applications for real-time monitoring and feedback [35,40,42,43,46,48,50,51,52]. Researchers utilized programming languages like Java and Xamarin for Android app development. One study extended their mobile app development to iOS platforms [51]. Mobile applications provided users with a convenient and portable means to interact with their health data, receive notifications, and manage their health on-the-go. One study used Blynk as the UI for their IoT system, providing mobile (both Android and iOS) and web app control [48]. Blynk enables real-time data visualization and interaction through customizable dashboards [54]. Six studies opted for web applications as their primary user interface, offering a platform-independent solution accessible through web browsers [36,37,38,41,44,53]. These UIs, developed using Python, JavaScript, Node.js, and PHP, enabled users to view comprehensive health data, interact with the system, and receive insights through a user-friendly dashboard. However, four studies did not develop any specific UI for users [39,45,47,49]. While the development of UIs did not inherently highlight accessibility, the use of mobile and web-based platforms facilitated real-time interaction and comprehensive data visualization. Researchers aimed to ensure effective interaction with health data, timely feedback, and efficient data management through these multi-platform approaches.

#### 3.3.4. Input Data and Classification

**Preprocessing:** The preprocessing steps varied across studies, with some providing detailed processes and others offering limited information. Data handling involved using ThingSpeak [35]. A 30 s push-up session was included for calibrating pressure sensors [38]. Denoising of sensor data using a Kalman filter was applied before feeding it into the ANN model [39]. Preprocessing steps included calibration against standard measures, conversion to JSON format, and publication to specific topics in a Redis process for further analysis [53]. Data normalization or scaling to 100 Hz for integration with body movement data was mentioned, though detailed steps were not exhaustively listed [42]. Real-time data processing and analysis based on threshold values were referenced without specifying preprocessing steps [40]. Data reduction using 2-D PCA for dimensionality reduction before classification was included [43]. Data were divided into different segments using window sizes of 2, 3, and 4 s for feature extraction [41].

**Feature Extraction and Preprocessing:** The reviewed studies employed various sensors and extracted a wide range of features for different healthcare applications. Motion-based features were commonly extracted using IMUs and accelerometers. These included angular rotation differences (delta x, y, and z), relative radial distance, angular distance [35], stride length, foot clearance, gait speed [39], and orientation angles [40]. More complex features such as signal amplitude area, synthetic acceleration, and time and frequency domain features were also extracted from activity sensors [43]. Some studies utilized advanced processing techniques, extracting features like mean values, standard deviations, entropy, cross-correlation, root mean square (RMS), zero-crossing, maximum values, and frequency-domain features from accelerometer and gyroscope data [41].

Physiological features were extracted using a diverse array of biosensors. EMG sensors provided data on the mean absolute value (MAV) and RMS of EMG signals [35]. CGM and fitness trackers were used to collect data on glucose sensor readings, heart rate, and stress levels [36]. More comprehensive physiological monitoring was achieved using devices like the Medical Blackbox, which measured pulse, blood oxygen levels, body temperature, and volatile organic compounds from breath and urine biochemistry [49].

Environmental features were also considered in several studies. These included ambient temperature, humidity, light, air quality, and the presence of flammable gases and smoke [49]. One study incorporated audio features such as energy, entropy, brightness, and formants [43]. Pressure-based features were extracted in some studies using FSR sensors. These included the mean and standard deviation of each channel, mean crossings, peaks, skewness, kurtosis, band power, mean frequency, and power spectrum of normalized weights [38]. In another study focusing on athletes’ health, heart rate monitors and temperature sensors were used to gather data on heart rate and body temperature [42].

Moreover, some studies use sensor data for regression analysis to calibrate and predict outcomes [53] or to classify normal and abnormal patient conditions [47], without specifying the exact features used. The choice of features varied depending on the specific application, ranging from fall detection [40] to classifying an athlete’s health state [42].

**Classification Models:** Logistic regression (LR), support vector machine (SVM), and k-nearest neighbors (k-NN) models were employed for the classification of different exercises [35]. Another study used k-NN to classify signs of mild cognitive impairment in older adults [49]. Long short-term memory (LSTM) neural networks were utilized to classify an athlete’s health state based on data such as heart rate, temperature, and humidity [42].

An artificial neural network (ANN) classifier was used to categorize gait patterns into in-toeing and out-toeing [39]. Threshold-based detection methods were employed to classify activities as fall events or non-fall events using sensor data [40]. Human physical activity recognition (HPAR) was focused on using classifiers such as KNN, naive Bayes (NB), random forest (RF), and Bayesian networks, utilizing data like movement patterns and physiological signals [41].

Some studies have explored classification in environmental and sports contexts. Air quality (AQ) levels were classified using logistic regression with audio, image, and environmental data inputs [43]. Hits to the boxing helmet were classified based on their strength and type, although the specific classification model was not mentioned [44]. Other research has employed a variety of classification models without specifying the target. Decision trees, random forests, SVM, k-NN, and ensemble methods were used for classification [38]. Sensor data were classified into normal patient conditions and various anomalies, though the specific classifier used was not specified [47]. Not all studies in this review focused on classification tasks. One study concentrated on data transmission without any classification tasks [46]. Another study did not specify any classification tasks or models [48]. Regression analysis was mentioned for calibration and prediction but did not provide details on classification targets or models [53].

#### 3.3.5. Data Exchange and System Security

**Data Exchange and System Latency:** Eighteen studies explored data exchange mechanisms within their proposed systems. Several studies leveraged cloud-based platforms. ThingSpeak and Blynk platforms were employed for exercise tracking and asthma care, respectively [35,48]. In another study, the Mi Fit smartwatch was used to save and transmit data [36]. Moreover, Bluetooth low energy (BLE) was a popular choice for real-time analysis and visualization across multiple studies [37,38,40,43]. To provide reliable bidirectional communication with minimal latency and efficient power management, the MiWi network protocol was used to transfer data from wristwatch sensors to gateways at 868 MHz [46]. Wi-Fi-based approaches were also common. Physiological data were transmitted to Android-based fog servers for processing and cloud storage via REST APIs [50,51]. Low-power Wi-Fi was used to transmit data from accelerometer and gyro sensors [39]. In another study, Wi-Fi using the MQTT protocol was used to transmit data from a Raspberry Pi-based sensing system [53]. Wireless sensor networks (WSNs) were employed in various configurations. Multi-hop WSNs were utilized for real-time monitoring of elderly individuals [52], while fault-tolerant WBANs were developed for physiological data collection and transmission [47]. Fog-centric IoT frameworks enabled timely health alert generation through wearable data management and analysis [42].

Throughput measurements and rates vary across the studies. In one study, the data sensing unit (DSU) samples and transfers sensing data via BLE at a rate of 5 Hz [38]. Throughput measurements for the study that used the MiWi protocol showed 25.84 kbps without the protocol, 19.13 kbps using it without security, 9.60 kbps with security enabled for transmission only, and 1.44 kbps with security for both transmission and reception [46]. High-throughput communication protocols ensured efficient data exchange between fog nodes and the cloud [42].

Latency details were also provided by several studies. In [35], the entire process from data acquisition to providing feedback through an Android application took approximately 700 ms over a 12 MB bandwidth internet connection. To improve latency, two studies leveraged fog computing to accelerate real-time data processing [42,50]. In [45], researchers have demonstrated the impact of user scale on system latency, reporting an increase from 0.62 min for 50 participants to 10.32 min for 800+ participants, highlighting the need for scalable system architectures to maintain performance under increasing loads [45].

**System Security:** Eight studies addressed security considerations [36,43,46,47,48,50,51,53]. The most robust measures were observed in [48], which implemented multiple security layers: Wi-Fi password protection, API keys and tokens, and SSL/TLS encryption to ensure secure data protection. Encryption was a commonly employed technique. Multiple studies utilized encryption to safeguard data transmission and storage [43,46,47,50,51,53].

Authentication was another critical component. Some studies incorporated secure authentication methods to control access to sensitive health information [47,51]. Network security was addressed through various protocols. MQTT with TLS encryption was used for secure data transmission in [53], while MiWi with encryption options was employed in [46].

Beyond security, several studies considered data privacy. One study described security services on their server platform, including API services for calibration and erase commands, but did not specify further security measures [53]. Another study verified secure bidirectional communication between devices, meeting minimum throughput requirements, while some studies mentioned privacy and security concerns but planned to address these in future iterations [46,51].

### 3.4. Reported Findings

Table 3 summarizes the reported outcomes, results, and conclusions from the reviewed studies. The studies demonstrate high accuracy in activity recognition, exceeding 94% for various classifiers [36,41]. Exercise recognition achieved 88% accuracy in person-dependent and 82% in person-independent scenarios [38], with specific exercises like T-bar and bicep curls recognized at 89% and 98% accuracy, respectively [35]. Health monitoring systems showed high usability, with system usability scale (SUS) scores of 83.0 [50,51]. Gait-pattern classification using ANNs reached 93% accuracy [39], while blood pressure measurements improved with root-mean-square errors of 9.76 mmHg for systolic and 5.56 mmHg for diastolic readings [53].

Fall detection systems performed exceptionally well, achieving 97.6% accuracy and 92.8% sensitivity [40]. Machine learning models for patients with mild cognitive impairment (MCI) showed accuracies ranging from 73.8% to 78% across different algorithms [49]. IoT-based interventions in diabetes management reduced HbA1c levels by 0.40% after 3 months and increased daily steps from 6743 to 8188 [37]. A COVID-19 symptom tracking system identified 11.97% suspected cases by week 6 and reduced social distancing violations by 3.61% [45].

Sensor validation studies confirmed the reliability of DS18B20 and MAX30100 sensors for real-time health monitoring [48]. An intelligent helmet system achieved 96% accuracy in detecting high-impact hits across 10 matches [44]. The MiWi network protocol provided reliable communication with a maximum throughput of 1.44 kbps for HR and SpO_2_ data transmission [46]. A WBAN-based fault-tolerant framework achieved 89% average accuracy in detecting faulty sensors [47]. Health zone and gym activity recognition (GAR) modules demonstrated over 97% accuracy in categorizing health zones and more than 89% accuracy in muscle group recognition [42]. Logistic regression achieved 85% accuracy in classifying Autism spectrum quotient (AQ) levels using audio, behavior, and environmental features [43]. An IoT-based health promotion system for the elderly achieved a 73% satisfaction rate among participants [52].

Many studies focused on improving classification model accuracy using techniques such as KNN, random forest, and feature fusion [35,36,38,39,43]. Research highlighted advancements in health monitoring and management systems, demonstrating effective support for the elderly, improved health outcomes, and efficient health zone classifications [37,42,48,49,50]. Enhancing system efficiency through using low-power sensors and achieving high-performance metrics in applications like fall detection were focal points [40,47]. Development and validation of prototypes showcased the applicability and effectiveness of the proposed systems in real-world scenarios [44,50,51,52]. Evaluating window sizes for feature extraction highlighted the importance of parameter tuning, showing significantly improved system performance [36,41]. Positive user acceptance and practical utility emphasized the feasibility and real-world applicability of the research [39,45,50].

### 3.5. Limitations

The review of 19 studies identifies several critical limitations affecting the generalizability, usability, and reliability of IoT-based health monitoring systems. Key issues include small sample sizes, which significantly limit the statistical power and generalizability of findings. Studies often included as few as five elderly users or four athletes, underscoring the urgent need for larger and more diverse sample populations to enhance the robustness of the results [35,36,37,38,39,40,41,42,43,44,46,48,50,52].

Wearable sensor discomfort and practicality issues also emerged as significant concerns. The design of wearable sensors, such as smart gloves or multiple sensors, often caused discomfort or hindered movement, impacting long-term use and data accuracy [35,38,43]. Moreover, the reliance on internet connectivity for real-time data transmission and cloud storage posed a major limitation, especially in areas with poor infrastructure. This dependence could significantly reduce the system’s usability and effectiveness in continuous health monitoring [39,40,41,43,44,46,48,50,51,53].

A recurring issue was the lack of extensive clinical validation, raising concerns about these findings in real-world clinical settings. Without rigorous testing in clinical environments, the effectiveness of these systems for broader health monitoring applications remains uncertain [35,37,38,39,40,43,47,48,49,50,51,52]. Additionally, many studies exhibit insufficient initial security measures to protect sensitive health data from advanced cyber threats. Enhanced security protocols are essential to ensure data privacy and safeguard against unauthorized access [47,48,50,51,53].

The scope of research in many studies was limited, often focusing on specific demographics or health parameters. This narrow focus necessitates broader studies to validate findings across diverse populations and health conditions [43,47,48,49,50,51]. Other notable limitations include the influence of environmental noise on data accuracy, which affected several studies. Noise interference impacts the accuracy of collected data and subsequent analysis, highlighting the need for advanced noise reduction techniques [38,39,43].

Several studies were limited to prototype models, indicating the need for future enhancements for broader application. These studies demonstrated initial feasibility but acknowledged that further development and rigorous testing are necessary to ensure practical usability and effectiveness [48]. User feedback highlighted areas for improvement in usability and design, emphasizing the need for more user-friendly interfaces and better overall system usability [51]. Specific activity focus in some studies, such as monitoring particular physical activities or health conditions, limited the applicability of the findings to broader contexts [37,41].

Battery life and device size posed practical challenges in some studies. These issues underscore the need for more efficient and compact system designs to enhance user comfort and data collection efficiency [53]. Additionally, the limited scope for qualitative insights and the potential for missing emerging studies highlight critical gaps that should be addressed in future research. Overall, these limitations highlight the critical need for larger sample sizes, extensive clinical validation, robust security measures, and practical, user-friendly sensor designs in future research to improve the reliability and applicability of IoT-based health monitoring systems.

### 3.6. Prospective Research Areas

The reviewed articles have suggested future research that could address several critical areas to enhance current IoT systems for user tracking. Direct cloud communication and wider connection ranges are among the key areas of focus. Researchers plan to implement direct cloud communication to streamline data processing and expand the connectivity range of their devices, which will support more comprehensive data collection and analysis [36,46]. Further development of smart fitness suites will include more comprehensive data collection and broader connectivity [35].

Long-term evaluations are essential to assess the sustained effectiveness and user acceptance of these systems. Plans for long-term evaluations aim to gather extensive data over prolonged periods, and extending study periods will help observe long-term effects and outcomes [37,50,51]. Increasing sample sizes and enhancing participant diversity are critical steps. Future studies will include more participants to improve the generalizability of findings, and involving a more diverse participant pool will enhance data collection [42,47]. Integrating developed systems into existing healthcare infrastructure is another key focus. Plans to integrate systems into hospital settings will enhance patient monitoring and care, and providing personalized health management recommendations indicates a move towards more tailored healthcare solutions [49,52]. Enhancing device and system capabilities is a priority, with a focus on miniaturization to improve user comfort and practicality. Testing system robustness in different environments will also be pursued [40,43,53].

Optimization of algorithms and incorporation of additional features are highlighted as future directions. Integrating optimization algorithms will improve system performance, and incorporating additional symptoms into tracking systems will provide more comprehensive health monitoring solutions [45,48]. Enhancing data security measures is crucial. Implementing more robust security measures will protect user data [51]. Expanding system capabilities and exploring new applications are important future directions. Plans to cover more scenarios and use cases will enhance system functionality. Using more sensors to capture a wider range of data will also be pursued [41,44]. Developing more specific models to address a broader range of scenarios will improve overall system applicability [39].

### 3.7. Future Study Recommendations

Building upon the existing research, we propose several novel directions for future studies that leverage cutting-edge technologies and concepts to address current limitations and emerging challenges in IoT-based health, wellness, and fitness tracking systems. At the heart of this evolution is the critical need for enhanced data security and privacy. Blockchain technology offers a decentralized approach to data integrity and sharing, potentially revolutionizing health data management [55]. Federated learning techniques present a promising avenue for privacy-preserving analytics, enabling machine learning models to be trained on diverse datasets without compromising individual privacy [56,57]. Future research should focus on developing comprehensive security measures that protect not only the data at rest and in transit but also the ML models themselves. Exploring advanced encryption techniques for secure data transmission and investigating methods to prevent adversarial attacks on AI models are crucial areas for further study [58]. Advanced AI techniques, including deep learning and reinforcement learning, should be explored for detecting subtle health anomalies and providing predictive analytics. Parallel research into explainable AI models is crucial to ensure transparency in health recommendations, fostering trust and user adherence [59]. Edge computing presents an opportunity to optimize real-time processing for health and fitness applications, enabling instantaneous analysis of critical health indicators. This aligns with the rollout of 5G technologies, promising high-bandwidth, low-latency connections for seamless monitoring and telemedicine [60]. As IoT health devices become ubiquitous, research into energy harvesting techniques and ultra-low power designs is essential to extend device longevity and reduce environmental impact [61,62]. Furthermore, future studies should prioritize the development of intuitive and accessible user interfaces for both end-users and system managers. Research into adaptive UI designs that cater to diverse user needs and abilities could significantly enhance user engagement and system effectiveness [63,64]. For system managers, investigations into dashboard designs that effectively communicate complex health data and system performance metrics are crucial for efficient monitoring and decision-making [65].

## 4. Conclusions

This systematic review demonstrates the transformative potential of multimodal IoT systems in enhancing health and wellness management through advanced M2M interactions. IoT technologies have revolutionized how we monitor, manage, and enhance user tracking by providing real-time data, cohesive system functionality, tailored user experiences, and personalized feedback. The analysis of sensors reveals significant variation in the types and configurations of sensors used for health monitoring. IMU sensors and motion detectors are highly effective for tracking physical activity and detecting postural changes, making them useful for fitness and rehabilitation applications. ECG and EMG sensors offer precise monitoring of cardiac and muscle activities, respectively. Studies collectively emphasize the importance of *multimodal sensor integration*, where combining data from various biosensors provides a comprehensive view of the user. For instance, combining heart rate monitors with motion sensors can improve the accuracy of activity recognition algorithms, leading to better health outcomes.

However, the review also reveals several critical limitations. Many studies suffer from small sample sizes, limiting the generalizability of their findings. There is a lack of comprehensive data integration, leading to fragmented insights rather than a holistic view of an individual’s health. Practical issues with wearable sensors, such as discomfort, limited battery life, and device size, hinder long-term usage and user compliance. The heavy reliance on internet connectivity poses challenges in areas with poor network infrastructure, leading to data loss or delays in processing. Security and privacy concerns are significant, with many studies lacking robust encryption and secure authentication protocols, exposing users to potential data breaches.

Future research must address these limitations by conducting studies with larger and more diverse sample populations. Researchers should focus on integrating comprehensive security measures to safeguard data and improve user trust. Additionally, developing more user-friendly interfaces and miniaturizing wearable sensors will enhance user comfort and system usability. Long-term evaluations are needed to assess the sustained effectiveness and user acceptance of these systems. Innovative technologies like AI and blockchain should be explored to enhance the accuracy, predictive capabilities, and security of IoT-based health monitoring systems. Incorporating multimodal sensor integration will provide a more comprehensive health monitoring solution. By addressing these areas, future research can optimize the implementation and integration of IoT technologies in health and wellness management, ultimately improving health outcomes and user engagement.

## Figures and Tables

**Figure 1 sensors-24-05939-f001:**
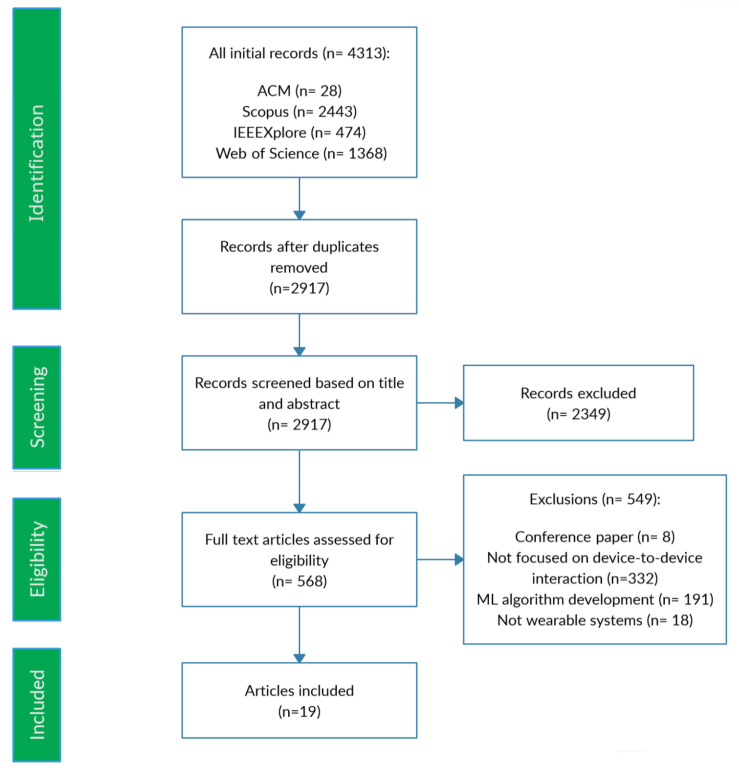
Flowchart of paper selection steps.

**Figure 2 sensors-24-05939-f002:**
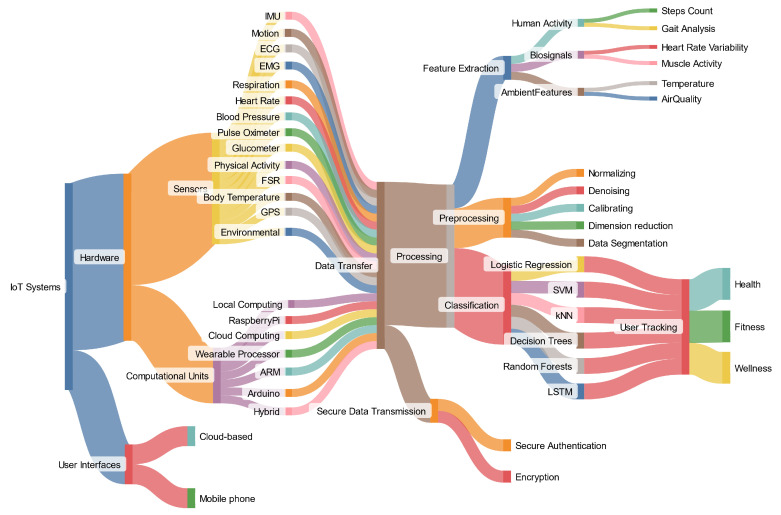
Alluvial diagram showcasing technologies and methods utilized in the included articles for IoT-based user-tracking systems.

**Table 3 sensors-24-05939-t003:** Summary of reported outcomes, results, and conclusions from reviewed studies.

Study	New Findings	Results	Conclusions
[35]	Validation using *t*-test: *p*-value < 0.02 (gyroscope), < 0.0001 (EMG)	KNN accuracy: 89% (T-bar), 98% (bicep curl)	Smart fitness suite for posture correction and exercise monitoring
[36]	Optimal 3 s window size for classification	Classification accuracy: >94% (RF, KNN), Mean glucose change: 7.4% (SD 1.5%)	Framework integrating wearable sensors and ML for health monitoring
[38]	A 3 s window size improves exercise recognition	Exercise recognition accuracy: 88% (person-dependent), 82% (person-independent), F-score: 0.889	Interaction between sensor data for accurate exercise recognition
[46]	MiWi network protocol for health data transmission	Communication throughput: 1.44 kbps (HR, SpO_2_ data)	Reliable health monitoring communication protocol
[49]	Cross-patient analysis for elderly people with MCI	Specificity/accuracy: kNN: 66.67%/76.01%; SVM: 63.34%/74%; RF: 64%/73.8%; LR: 65%/78%	Practical benefits for elderly care with MCI
[50]	High usability score on SUS	SUS score: 83.0	Reliable and user-friendly health monitoring system
[51]	Real-time monitoring and alerts in Abuelómetro e-health system	SUS score: 83.0, encrypted data transmission	Secure and effective health data management
[39]	ANN-based gait-pattern classification	Accuracy: 93% (in-toeing, normal, out-toeing)	Potential of ANN in gait-pattern classification
[53]	AdaBoost-enhanced DT and SVM improve BP measurement accuracy	Systolic RMSE: 9.76 mmHg, diastolic RMSE: 5.56 mmHg	Advancements in BP measurement accuracy using advanced algorithms
[52]	IoT-based health promotion system for elderly	Satisfaction rate: 73%, assistance needed: 31%	Positive impact on health promotion and user satisfaction
[48]	Validation of DS18B20 and MAX30100 sensors	Real-time data visualization, stable parameters	Effective sensor validation and real-time monitoring
[37]	Exercise interventions reduce HbA1c levels and increase steps	HbA1c: −0.40% (3 months), −0.19% (6 months), Steps: 6743 to 8188	Health benefits of exercise interventions in diabetes management
[47]	WBAN-based fault-tolerant framework	Accuracy: 89%, specificity: 74% (faulty), 91% (normal), 80.8% (sick), 100% (some cases)	Robust framework for reliable sensor data classification
[42]	Health zone and GAR modules for health zone and muscle group modeling	Accuracy: >97% (health zones), >89% (muscle groups), >80% (overall)	Effective modular systems in health and exercise monitoring
[40]	Fall detection system using various models	Accuracy: 97.6%, sensitivity: 92.8%, specificity: 100%, fall types: 100% (forward/backward), 92% (right), 88% (left), 84% (sitting)	High-performance fall detection with detailed metrics
[43]	Logistic regression for Autism Spectrum Quotient (AQ) levels	Accuracy: 85%, significant correlations with AQ scores	Potential of logistic regression in predicting autism spectrum levels
[44]	IoT-based intelligent helmet for high-impact hits	Accuracy: 96% across 10 matches	Practical applicability in sports safety monitoring
[45]	Smart COVID-Shield for symptom tracking and social distancing	Suspected COVID-19: 11.97% (week 6), social distancing violations: −3.61%, latency: 10.32 min (800 participants)	Effective COVID-19 symptom tracking and management
[41]	WIoT platform for HPAR with high classifier accuracy	Accuracy: >94% (RF, DT, rule-based), >96% (specific activities), 10-fold cross-validation	Effectiveness of WIoT platform in health and activity recognition

Acronyms: ANN: artificial neural network, BP: blood pressure, DT: decision tree, EMG: electromyography, F-score: F-Measure, GAR: gym activity recognition, HbA1c: hemoglobin A1c, HPAR: human physical activity recognition, IoT: Internet of Things, kNN: k-Nearest Neighbors, KNN: K-Nearest Neighbors, LR: logistic regression, MCI: mild cognitive impairment, MiWi: microchip wireless, ML: machine learning, RF: random forest, RMSE: root-mean-square error, SD: standard deviation, SpO_2_: oxygen saturation, SUS: system usability scale, SVM: support vector machine, WBAN: wireless body area network, WIoT: wireless Internet of Things.

## Data Availability

Data supporting reported results can be accessed by contacting the corresponding author.

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
