# Peer review of "Systematic Review of IoT-Based Solutions for User Tracking: Towards Smarter Lifestyle, Wellness and Health Management"

_sensors, 2024, doi:10.3390/s24185939_

Round 1
Reviewer 1 Report
Comments and Suggestions for Authors
The publication "Systematic Review of IoT-Based Solutions for User Tracking: Towards Smarter Lifestyle,Wellness, and Health Management" is well written, easy to read and understand. Table 1 orientation is not helpful for reading, I would recommend to either split it in two tables or reduce the amount of information, so it fits with portrait orientation.
Furthermore, the application of PRISMA methodology in the systematic review of IoT in health and lifestyle management provides several strengths, including structured reporting, comprehensive search strategies, and quality assessment.
Critical limitations should be considered as well e.g. limited scope for qualitative insights, and the potential for missing emerging studies.
To get the best of PRISMA in this context, future reviews should consider supplementing quantitative data with more qualitative insights .
Author Response
Hello,
Thank you so much for your comments. Here we provide our responses to your comments.
Comment: Table 1 orientation is not helpful for reading, I would recommend to either split it in two tables or reduce the amount of information, so it fits with portrait orientation.
Response: Thank you for your suggestion. Now in the revised text we have fitted table 1 into the portrait orientation to increase readability.
Comment: Critical limitations should be considered as well e.g. limited scope for qualitative insights, and the potential for missing emerging studies.
Response: We revised the limitation section and included mentioned points.
Comment: To get the best of PRISMA in this context, future reviews should consider supplementing quantitative data with more qualitative insights .
Response: Thank you so much for your feedback. We included a PICOS table to support PRISMA in a qualitative way and we included more insights in relevant sections to support PRISMA results quantitatively.

Reviewer 2 Report
Comments and Suggestions for Authors
There is no comparison with other systematic mappings, or it is not made clear to the reader. What is the unique scientific contribution of this paper that distinguishes it from existing mappings?
Why were only four article databases used? For example, Springer contains a significant number of articles related to this subject.
Section 2 ends abruptly while explaining the methodology, and only in the discussion does the reader learn how many articles were found.
In sections 3.1 to 3.4, the authors mix the methodology with the results and then revert to discussing the methodology. This structure confuses the reader.
Is there any particular reason for choosing only papers from 2019 to 2023?
Unfortunately, the paper does not have the text quality required for acceptance in its current form.
Comments on the Quality of English Languagefew typos only.
Author Response
Hello,
Thank you so much for your comments. Here we provide our responses to your comments.
Comment: There is no comparison with other systematic mappings, or it is not made clear to the reader. What is the unique scientific contribution of this paper that distinguishes it from existing mappings?
Response: Thank you for this comment. We revised the introduction and pointed out the novelty of our approach to surveying and the insights that we provide differing from existing mappings.
Comment: Why were only four article databases used? For example, Springer contains a significant number of articles related to this subject.
Response: In this work, we leveraged four main indexing databases. The Springer journals are indexed by Scopus and Web of Science [1] that we used. In our searches, we prioritized databases that provide comprehensive and overlapping indexing of relevant journals, Springer included, to optimize search efficiency and ensure a focused review process.
[1] https://www.springeropen.com/get-published/indexing-archiving-and-access-to-data/new-content-item
Comment: Section 2 ends abruptly while explaining the methodology, and only in the discussion does the reader learn how many articles were found.
Response: Thank you very much for this comment. Section 2 (Methodology) outlines our approach, while the "Results and Discussion" section presents the outcomes. This structure ensures a clear separation between methods and findings, as is standard required by this journal. To improve readability and provide clarity on the number of articles included, we have added a “Study Selection Process” subsection to the methodology section, where we outline the article counts and selection details.
Comment: In sections 3.1 to 3.4, the authors mix the methodology with the results and then revert to discussing the methodology. This structure confuses the reader.
Response: Thank you for highlighting this issue. The term “methodology” in the section titles refers to two distinct aspects: our methodology for conducting the literature review and the methodologies used by the papers we reviewed. To improve clarity, we have revised the section titles to better differentiate between our review methods and the methodologies employed in the studies.
Comment: Is there any particular reason for choosing only papers from 2019 to 2023?
Response: We have chosen to include studies from the last 5 years (2019 to 2023 inclusive) to include the most recent and relevant research to modern IoT realizations.
Comment: Unfortunately, the paper does not have the text quality required for acceptance in its current form.
Response: We appreciate your feedback. We have manually proofread and passed the text through two grammar and style checkers (Grammarly and Microsoft Office). We have further reviewed and refined the manuscript to ensure clarity and coherence.

Round 2
Reviewer 2 Report
Comments and Suggestions for Authors
The authors solved all the issues that were pointed out previously, no more issues from my side.